# Molecular Mechanisms of Prostate Cancer Development in the Precision Medicine Era: A Comprehensive Review

**DOI:** 10.3390/cancers16030523

**Published:** 2024-01-25

**Authors:** Shigekatsu Maekawa, Ryo Takata, Wataru Obara

**Affiliations:** Department of Urology, Iwate Medical University, Iwate 028-3694, Japan; rtakata@iwate-med.ac.jp (R.T.); watao@iwate-med.ac.jp (W.O.)

**Keywords:** prostate cancer, next-generation sequencing, mutagenesis, signal pathways, cancer immunology, epigenomics, non-coding RNAs, drug resistance, poly (ADP-ribose) polymerase inhibitors, immune checkpoint inhibitor

## Abstract

**Simple Summary:**

Prostate cancer (PCa) is characterized by various genomic alterations that play a pivotal role in carcinogenesis. Efforts in precision medicine aimed at improving diagnosis, prevention, and surveillance based on genetic alterations are advancing. Notably, no tumor markers surpass prostate-specific antigen in specificity, and existing treatments primarily target the androgen receptor axis, with exceptions for patients with alterations in homologous recombination repair-related genes, such as *BRCA1/2* and *ATM*, who may benefit from poly (ADP-ribose) polymerase inhibitors. In order to delineate the current state of research on PCa, we provide an overview of cutting-edge genomic research, including genome alterations, cancer immunology, and non-coding RNAs, in PCa. These aspects are relevant to comprehending the molecular mechanisms underlying health disparities, PCa initiation and progression, and drug resistance in the precision medicine era.

**Abstract:**

The progression of prostate cancer (PCa) relies on the activation of the androgen receptor (AR) by androgens. Despite efforts to block this pathway through androgen deprivation therapy, resistance can occur through several mechanisms, including the abnormal activation of AR, resulting in castration-resistant PCa following the introduction of treatment. Mutations, amplifications, and splicing variants in AR-related genes have garnered attention in this regard. Furthermore, recent large-scale next-generation sequencing analysis has revealed the critical roles of AR and AR-related genes, as well as the DNA repair, PI3K, and cell cycle pathways, in the onset and progression of PCa. Moreover, research on epigenomics and microRNA has increasingly become popular; however, it has not translated into the development of effective therapeutic strategies. Additionally, treatments targeting homologous recombination repair mutations and the PI3K/Akt pathway have been developed and are increasingly accessible, and multiple clinical trials have investigated the efficacy of immune checkpoint inhibitors. In this comprehensive review, we outline the status of PCa research in genomics and briefly explore potential future developments in the field of epigenetic modifications and microRNAs.

## 1. Introduction

Prostate cancer (PCa) ranks as the fourth most commonly diagnosed cancer globally, accounting for 7.3% of all cases. In 2020, PCa held the position of the second most prevalent cancer (at 14.1%) and the fifth leading cause of cancer-related mortality (at 6.8%) among men [1]. Factors such as advanced age, African descent, and a family history of the disease are associated with poorer prognoses for PCa [2].

Age-adjusted incidence rates for PCa in Europe, America, and Oceania range from 59 to 73 cases per 100,000 men, surpassing those in Africa and Asia, where rates are 29.7 and 13.6 cases per 100,000 men, respectively. However, age-standardized mortality rates are highest in Africa (at 16.3 cases per 100,000 men), followed by Latin America, the Caribbean, Europe, Oceania, Northern America, and Asia (with rates of 4.1–14.2 per 100,000 men) [3].

Treatment for PCa primarily involves androgen deprivation therapy (ADT), androgen receptor (AR)-signaling inhibitors (ARSIs), and chemotherapy. Following initial hormone therapy, metastatic castration-sensitive PCa (mCSPC) often progresses to metastatic castration-resistant PCa (mCRPC) within approximately 1–2 years. The median survival time and cancer-specific mortality for mCRPC are 22 months (95% confidence interval [CI]: 21.0–22.9) and 30 months (95% CI: 28.3–31.7), respectively [4]. In one study, the 1- and 5-year survival rates for mCRPC were estimated at 58, 37, and 58% and 17, 4, and 9% in German, Swedish, and US cohorts, respectively [5]. In another study, the median overall survival (OS) for chemotherapy-naïve mCRPC was significantly longer in the ARSI group (treated with abiraterone acetate or enzalutamide; 34.7 and 35.3 months, respectively) than in the placebo group [6,7]. In Japanese patients with mCRPC, the median progression-free survival (PFS) and OS based on prostate-specific antigen (PSA) were 5.6 (95% CI: 4.7–6.8) months and 26.1 (95% CI: 22.1–30.5) months, respectively [8]. The median OS of diagnosed CRPC in patients with mCRPC or non-metastatic CRPC (m0CRPC) was 29.2 (95% CI: 15.3–37.4) months and 41.7 (95% CI: 16.1–not estimable) months, respectively, with no discernible difference between de novo (28.1 months; 95% CI: 15.3–38.2) and metachronous (33.3 months; 95% CI: 14.8–65.9) mCRPC at Iwate Medical University Hospital (our data).

Recently, precision medicine targeting driver genes, such as treatment employing poly (ADP-ribose) polymerase (PARP) inhibitors for *BRCA* mutations identified through companion diagnostics, has been realized. Next-generation sequencing (NGS) technology has facilitated comprehensive and exhaustive genome analysis, revealing a multitude of genetic mutations associated with cancer initiation and progression. Furthermore, NGS has enabled routine clinical examination of individual genetic abnormalities. Consequently, personalized treatments based on genetic anomalies have been developed, even in the context of PCa, with accumulating evidence supporting this approach. Recent reports have highlighted the utility of Decipher Prostate testing in determining the treatment approach for recurrence after radical prostatectomy [9]. Despite extensive genome-wide sequencing studies on PCa, no biomarker has surpassed PSA for several decades.

Therefore, in this review, we aim to provide an overview of recent advances in NGS findings, potential biomarkers for various treatments, and the current status and challenges of precision medicine for PCa based on genetic anomalies.

## 2. Genomic Alterations

In PCa, the involvement of driver genes is highly diverse in terms of both carcinogenesis and disease progression. This diversity varies significantly among different cases as well as across various stages of PCa. Considering the variation in driver genes, precision medicine assumes profound significance in the context of PCa (Table 1). The most prevalent genetic anomalies observed in PCa include point mutations in genes such as *SPOP*, *FOXA1*, and *TP53*, as well as copy number alterations (CNAs) in *AR*, *MYC*, *RB1*, *PTEN*, *CHD1*, and fusion genes associated with the *ETS* (E26 transformation-specific) family [10,11,12,13,14,15,16,17]. Alterations in pathways involving *AR*, *TP53*, the cell cycle, and MYC are strongly associated with the development of castration resistance and OS. By contrast, alterations in the Wnt pathway and *SPOP* mutations are indicative of a more favorable prognosis [18,19,20,21,22,23]. Notably, changes in *AR* and PI3K/Akt pathway components appear to result from continuous systemic therapy [24,25]. Conversely, PTEN loss, *SPOP* mutations, and alterations in DNA damage-repair mechanisms, including homologous recombination repair (HRR) and mismatch repair (MMR), tend to manifest early in prostate tumorigenesis, prior to metastatic progression, and generally remain stable over time [24,26].

The prevalence of specific alterations varies in different clinical contexts. For instance, alterations in *TP53* (35% vs. 29%), *BRCA2* (10% vs. 4%), *PIK3CA* (8% vs. 2%), *RB1* (7% vs. 3%), and *APC* (11% vs. 9%) are more frequent in high-volume diseases than in low-volume diseases. Alterations in *CDK12*, *PTEN*, and *FOXA1* are similarly detected between CSPC and CRPC, whereas those in *ATM* and *RB1* exhibit a similar prevalence between de novo and recurrent metastatic disease, with *CDK12* (6% vs. 1%) and *FOXA1* (17% vs. 10%) alterations more commonly observed in the de novo metastatic state. The PI3K/Akt pathway is affected in 35–40% of mCSPC cases [18,19,20,21,22,23,28,31,32,33,34]. In mCSPC, *TP53* and *PTEN* mutations, or CNAs, have been observed in 32% and 20% of cases, respectively [35].

In mCRPC, the most frequently aberrant genes include *AR* (62.7%), the ETS family (56.7%), *TP53* (53.3%), and *PTEN* (40.7%). The frequency of alterations in these genes is significantly higher than that observed in localized cancer [36]. In mCRPC, non-AR-related clinically actionable alterations include aberrations in the PI3K/Akt pathway (49%), DNA repair pathway (19%), RAF kinases (3%), CDK inhibitors (7%), and Wnt pathway (5%) [36]. Notably, alterations in *AR*, *TP53*, *RB1*, *PTEN*, and *ATM* are enriched in mCRPC when compared with those in mCSPC [28].

Large-scale genome profiling targeting Asian populations has revealed genetic abnormalities (such as those in *FOXA1*, *ZNF292*, and *CHD1*) that differ in frequency from those previously reported in Caucasian populations. This suggests potential variation in driver genes for PCa among different ethnicities. Conversely, *AR* abnormalities are relatively rare in localized PCa, irrespective of ethnicity [37].

### 2.1. Fusion Genes from the ETS Family

Among the most prevalent genomic alterations in PCa are fusions involving androgen-regulated promoters with ERG and other members of the ETS family of transcription factors. ETS fusions are observed in over 50% of cases [10,36,38,39]. In particular, the TMPRSS2–ERG fusion represents the most frequent molecular alteration in PCa [40], occurring in 40–50% cases, equivalent to >100,000 cases annually in the United States alone [41]. Notably, Asian patients with PCa exhibit fewer TMPRSS2–ERG fusions than do Caucasian or African-American patients [42,43,44]. TMPRSS2 expression is upregulated by androgenic hormones in PCa cells but downregulated in androgen-independent PCa tissue [44]. Genetic polymorphisms associated with PCa susceptibility may vary by race and ethnicity. For instance, Takata et al. [45] identified five SNPs (rs13385191, rs12653946, rs1983891, rs339331, and rs9600079) linked to PCa susceptibility through a genome-wide association study of a Japanese population. However, only two SNPs (rs13385191 and rs1983891) were associated with PCa susceptibility in a Latino population [46]. The TMPRSS2 Met160Val polymorphism is a genetic risk factor for sporadic PCa in the Japanese population [44]. In the early stages of PCa, frequent genomic alterations include TMPRSS2–ERG fusion in 40–60% of patients [10,47]. The presence of TMPRSS2–ERG in PCa has been associated with more aggressive biological behavior and a poorer prognosis [48,49].

### 2.2. AR and AR-Related Genes

The dysregulation of AR and AR pathway-related genes plays a central role in PCa pathogenesis. Abnormalities in the AR pathway include gene amplifications, mutations, splicing variants, and ligand-independent AR activity. Furthermore, these abnormalities can result from coding regions as well as AR enhancer region amplifications and genomic structural rearrangements capable of producing constitutively active truncated AR isoforms, notably AR splice variant 7 (AR-V7) [50].

Over 70% of PCa cases exhibit AR pathway aberrations, including those involving *SPOP* or *FOXA1*, with the majority resulting from direct alterations affecting AR through amplifications and mutations [36]. Alterations in androgen receptor signaling are more frequently observed in metastatic samples, with AR amplification or mutation being the most common alteration (50–80%). *SPOP* mutations, while present, are relatively less frequent in metastatic samples [10,36,51,52]. AR alterations are rarely observed prior to ADT, although they are enriched in patients with mCRPC who have a higher disease burden and prior exposure to ARSIs [53]. By contrast, changes in *AR* copy number status are not evident in patients treated with docetaxel or cabazitaxel [54,55]. Additionally, truncated AR splice variants were previously believed to be primarily expressed in mCRPC, and the presence of AR-V7 was associated with resistance to hormone therapy and taxane chemotherapy [56,57]. The detection of *AR* gene amplification in plasma cell-free DNA has consistently been linked to shorter PFS and OS in patients treated with ARSIs [58,59,60]. In a past study, the AR activity of tumors was assessed by calculating the AR activity score based on the expression patterns of 20 genes that were experimentally validated as AR transcriptional targets. This scoring indicated a wide range of AR activity between PCa cases and among different genomic subtypes [61].

SPOP plays a role in protein degradation by acting as a substrate adaptor, enabling ubiquitination by the Cullin3–RING complex [62]. In PCa, *SPOP* functions as a tumor-suppressor gene by promoting the degradation of multiple oncogenic substrates, including AR and ERG [63,64]. *SPOP* mutations represent the most common point mutations in localized PCa [10,12]. The frequency of *SPOP* mutations in localized disease (10–15%) and metastatic PCa (5–10%) aligns with the observation of a more favorable disease prognosis in de novo mCSPC [23]. Alterations in SPOP are enriched in locoregional and early-stage disease; therefore, tumors with *SPOP* mutations may exhibit increased sensitivity to ADT [12,26,28]. Furthermore, the potential of *SPOP* mutations as predictive biomarkers may manifest in enhanced sensitivity to second-generation ARSIs and taxane chemotherapy in mCRPC [65].

### 2.3. TP53

When comparing clonal alterations between a matched primary localized tumor and a subsequent metastatic tumor from the same patient, tumors acquired at a later stage typically exhibit a higher mutation count than do tumors acquired at an earlier stage. *TP53* alterations emerge early in affected patients and are present in all tumors from the same patient. *TP53* mutations are clonal, even in cases of both primary localized and subsequent metastatic PCa [28]. A high frequency of *TP53* alterations is observed in primary localized tumors, consistent with prior reports of aggressive behavior of PCa caused by *TP53* alterations [66,67,68]. The timing of *TP53* alterations in tumors, specifically whether these alterations are present in tumors early in their evolution or acquired later during disease progression, remains uncertain. Somatic alterations in BRCA2 have been identified in matched tumors, suggesting that somatic loss-of-function alterations in this gene occur early in tumorigenesis. By contrast, alterations in AR do not occur early in matched samples, which is consistent with treatment-related changes that promote castration resistance [28].

### 2.4. Mitogen-Activated Protein Kinase (MAPK) Pathway

Genes within the MAPK pathway are affected in 25% of tumors [10]. The presence of RAF kinase fusions in 3% of individuals with mCRPC suggests the potential utility of pan-RAF inhibitors or MEK inhibitors [36].

### 2.5. PI3K/Akt Pathway

Alterations in the PI3K/Akt pathway constitute the second most frequently observed pathway aberration in metastatic PCa [10,36,69,70]. The PI3K/Akt pathway may be antagonized by several phosphatases, including PTEN and PHLPP. PI3K activation results in the phosphorylation of AKT and its downstream genes, including the mammalian target of rapamycin (*mTOR*). Phosphorylated AKT is an indicator of PI3K/Akt pathway activation [71,72], and AKT activates nuclear factor-κB (NF-κB) signaling in CRPC, with levels of NF-κB being higher in castration-resistant cell lines than in androgen-dependent cell lines [73]. PTEN, which leads to the downstream activation of the PI3K/Akt and AR signaling pathways, plays a crucial role in tumor suppression, and its loss is associated with poorer clinical outcomes in patients with mCRPC [18,53,74,75]. Considering that PTEN-deleted tumors may rely on PIK3CB owing to feedback inhibition from PIK3CA, the simultaneous loss and mutation of PTEN and PIK3CB could result in increased PI3K/Akt pathway activity. This observation suggests the existence of a subset of tumors for which a combination of PI3K and androgen signaling inhibition might be effective. In the context of the PI3K/Akt pathway, specific inhibitors targeting PIK3CB may be beneficial for individuals with mutations, amplifications, and/or fusions involving the related gene. Notably, some individuals who demonstrate sustained responses (lasting more than 1 year) to PIK3CB-specific inhibition possess activating mutations or amplifications in *PIK3CB* [27]. PTEN loss occurs in 30–50% of PCa cases [36,51,52] and is recognized as an early genomic event in PCa initiation [10,76]. However, heterogeneity in clonal composition is evident in castration-naïve tumors [77]. In primary PCa, the deletion or mutation of *PTEN* is observed in 17% of cases [10]. In mCRPC, biallelic loss and hotspot mutations of *PTEN* (40–60%), as well as amplifications and activating fusions in *PIK3CA* (5%) and *AKT1* (1%), are observed [15,36,78].

### 2.6. NF-κB Signaling

The NF-κB/Rel proteins include NF-κB2 p52/p100, NF-κB1 p50/p105, c-Rel, RelA/p65, RelB, and others. These proteins function as dimeric transcription factors, primarily regulating the transcription of genes involved in a wide range of biological processes such as apoptosis, angiogenesis, inflammation response, cell survival, inflammation, stress response, B-cell development, and lymphoid tissue formation [79,80,81,82,83,84].

The activation of NF-κB (p65) and Sonic Hedgehog pathways in PCa is positively correlated, and the overexpression of NF-κB(p65), SHh, and GLI1 is observed in aggressive PCa tissue cores featuring a high Gleason score and advanced clinical stage. These pathways play key roles in the progression of advanced PCa and are potential targets for therapeutic intervention [85].

The involvement of NF-κB signaling in the onset and maintenance of CRPC is considerable, irrespective of whether it is due to abnormal AR activity or AR-independent mechanisms [86]. Moreover, NF-κB plays a crucial role in the development of treatment resistance and metastasis in PCa. Specifically, the RelB-activated noncanonical NF-κB pathway could prolong and enhance NF-κB activity [87]. In addition, the NF-κB complex can promote the survival of tumor cells in PCa through anti-apoptotic signaling. The activation of NF-κB is sustained by a positive feedback loop. Such activation is associated with PCa progression to a metastatic phenotype and mCRPC. A novel role of NF-κB kinase (IKK)-α in the progression of NF-κB-independent PCa and its progression to CRPC has been elucidated, revealing a mechanistic link between inflammation and PCa [88]. Additionally, NF-κB is implicated in the activation of the PI3K/Akt pathway and some non-coding RNAs, contributing to PCa progression [73,89,90,91].

### 2.7. Cell Cycle Pathway

RB1 loss is detected in 21% of PCa cases [36]. Focal amplifications involving *CCND1* (9%), as well as less frequent (<5%) events affecting *CDKN2A*/*B*, *CDKN1B*, and *CDK4*, are observed in mCRPC [36]. Disruptions in the cell cycle, such as *CCND1* amplification or CDKN2A/B loss, may lead to an increased responsiveness to CDK4 inhibitors, as observed in other tumor types [27]. Pre-clinical findings indicate that selectively targeting CDK4/6 represents a therapeutic approach for both early-stage and advanced prostate PCa [92]. Mutations or deletions of *TP53* and *RB1* are more prevalent in metastatic than in localized PCa [10,36], and these alterations are associated with a poorer clinical outcome and aggressive behavior [35,66,67,93,94].

### 2.8. DNA Repair Pathway

Alterations in the DNA repair pathway are more frequent in mCRPC than in localized PCa or CSPC [10,36]. Genetic abnormalities associated with the DNA damage response (DDR) are observed in approximately 23–28% of mCRPC cases, including both germline and somatic variants [10,11,20,28,29,30].

#### 2.8.1. HRR Genes

Among common gene abnormalities, *BRCA2* and *ATM* mutations constitute a significant majority [28]. In advanced PCa, *BRCA2* (7–13%), *BRCA1* (0–1%), *ATM* (5–7%), and *CDK12* (4–6%) exhibit frequent mutations [36,51,52,95]. The prevalence of pathological variants in *BRCA1/2* and *ATM* varies across different stages of PCa (Table 1) [10,36,76,96]. *ATM* alterations are associated with a poor prognosis; germline mutations are linked to higher tumor grades and lethal PCa [97,98,99]. *CDK12* alterations are associated with aggressive clinical features and a poor prognosis in mCRPC [100]. In a study of PCa patients who underwent radical prostatectomy or localized radiation therapy, both metastasis-free survival and disease-specific survival were significantly shorter in patients with *BRCA* mutations than in non-carriers [101]. In localized PCa, *BRCA2* germline mutations are associated with early-onset, high-grade tumors and a poor prognosis [101,102]. *BRCA2* alterations are enriched in mCRPC relative to their frequency in mCSPC [35]. Notably, approximately half of these pathological variants are attributed to germline mutations, and they are frequently accompanied by evidence of tumor loss-of-heterozygosity [28,103] as well as characteristic patterns of genomic and epigenomic instability [104]. The presence of germline *BRCA2* alterations in primary PCa is associated with poor survival outcomes [105]. In addition, Japanese patients with mCRPC-causing germline *BRCA1/2* mutations exhibit a significantly poorer prognosis and PFS following ARSI and taxane chemotherapy than non-careers [106]. *BRCA2* germline and/or somatic mutations are linked to aggressive pathologic features such as intraductal carcinoma of the prostate [107,108], and patients with these mutations appear to exhibit a poorer response to ARSIs than those seen in patients with wild-type *BRCA2* [53,109]. Regardless of patient ethnicity, PCa with germline or somatic HRR abnormalities exhibits increased aggressiveness, leading to enhanced resistance to ARSI and resulting in poor OS for CRPC [8,105,109,110,111,112,113]. Furthermore, the OS of patients with CRPC who underwent ARSI or were administered olaparib varies depending on the specific mutations within HRR-related genes. Therefore, personalized treatment recommendations are likely to be based on the results of genomic profiling (Table 2).

#### 2.8.2. MMR Genes and Microsatellite Instability (MSI)

Tumors with MMR gene deficiencies or MSI often exhibit an augmented antitumor immune response characterized by an increased density of tumor-infiltrating lymphocytes (TILs) [114,115]. This phenomenon is attributed to elevated mutation rates and enhanced levels of neoantigens in MMR-deficient tumors. Such neoantigens arise through various mechanisms, including mutant peptides, frameshift mutations, and insertions or deletions (indels) in coding microsatellites [116,117]. These neoantigens are presented on the cell surface by MHC-I molecules, facilitating the T-cell-mediated elimination of tumor cells. Approximately 3–8% of PCa cases, with a germline mutation rate of approximately 1%, are linked to the deficiency of MMR genes, including *MSH2*, *MSH6*, *PMS2*, and *MLH1*. Such deficiency leads to hypermutation, MSI, and characteristic trinucleotide mutational signatures [28,36,118,119,120,121,122,123]. MMR genes play crucial roles in the recognition and repair of incorrect insertions, deletions, and base misincorporations during DNA replication or recombination [124]. Among them, germline mutations in genes associated with Lynch syndrome (21.9%) have been observed [118]. Hypermutation occurs in 12% of advanced PCa cases, all of which exhibit mutations in MMR genes and exhibit MSI. Such mutations often involve complex structural rearrangements in *MSH2* or *MSH6* rather than epigenetic silencing of *MLH1* [122]. Although MMR protein loss and MSI are evident in both primary and advanced PCa, only a few MMR mutations have been identified [125,126,127,128,129,130,131,132]. Mutations in MMR genes in PCa are strongly associated with elevated neoantigen expression and TIL accumulation [133].

## 3. Immune Environment

In the context of PCa, the immune environment and hormone therapy involve macrophages/antigen-presenting cells (APCs), CD8-positive cytotoxic T lymphocytes, CD4-positive helper T lymphocytes, and natural killer (NK) cells, all of which recognize and target cancer cells, contributing to antitumor immunity. Several studies have reported on the expression of these cells within prostate tissues; notably, fewer TILs have been identified in poorly differentiated PCa than in benign prostatic hyperplasia, suggesting a potential link between immune cell deficiency and cancer progression. The infiltration of these immune cells within the tumor microenvironment of PCa tissue has also been associated with prognosis; patients with PCa and a higher number of infiltrating TILs tend to have better outcomes than those with fewer infiltrating TILs [134,135,136]. The examination of CD68-positive tumor-infiltrating macrophages (TAMs) in radical prostatectomy specimens of patients with PCa revealed that cases with a higher number of infiltrating cells were associated with a significantly worse prognosis for PSA recurrence than those with fewer infiltrating cells [137]. In PCa, an efficient tumor immune response is hindered by TILs that are either inactive or belong to the immunosuppressive T regulatory cell (Treg) subtype. Tregs represent subgroups of CD4+CD25+ and CD8+/FoxP3+ T lymphocytes with the ability to directly suppress immune responses. These Tregs are negatively correlated with prognosis in patients with PCa, as they contribute to cancer progression [138,139,140]. ADT is associated with the regulation of the immune system, i.e., the control of male hormones can either promote thymus formation and B-cell proliferation or suppress the tumor-infiltrating Tregs and can induce NK cell infiltration and CD4+ T-cell infiltration within PCa tissues. This regulation of the immune response in PCa may enhance the antitumor effects through the combination of hormone therapy and immunotherapy [141,142,143,144].

## 4. Epigenetic Modifications

Epigenetic traits represent inheritable characteristics resulting from chromosomal alterations or DNA modifications without alteration of the DNA sequence [145]. Epigenetic modifications, including acetylation, methylation, ubiquitination, and phosphorylation, play pivotal roles in transcription, DNA repair, and replication [146]. Epigenetic regulation is a dynamic and reversible process involving the addition of epigenetic marks to histones or DNA, the recognition or recruitment of these marks, and the removal of these marks by epigenetic writers, readers, and erasers, respectively.

PCa is associated with alterations in DNA methylation, histone modification, chromatin accessibility, and 3D chromatin structure. Such changes contribute to disease development and progression by influencing gene expression, promoting cellular plasticity, and facilitating CRPC emergence under treatment pressure. The dysregulation of chromatin and epigenetics confers the complete range of cancer hallmarks by altering transcriptional regulation, mediating all defined hallmarks of cancer, and promoting cancer cell plasticity [147].

Aberrant histone modifications can lead to the upregulation of oncogenes or the suppression of tumor-suppressor genes. Notably, histone methylation/acetylation and DNA methylation centrally control gene expression, thereby promoting the progression and metastasis of PCa [148,149,150,151,152]. Chromatin regulators such as EZH2 play a critical role in driving epigenetic reprogramming in PCa, being overexpressed in PCa and significantly overexpressed in CRPC [153,154,155]. Bromodomain (BRD)-containing proteins serve as chromatin readers, recognizing mono-acetylated histones and thereby initiating chromatin remodeling and transcription. Dysregulation of BRDs is a common occurrence in cancer, including metastatic PCa [156]. Within the BRD family, BRD4 stands out as an important co-regulator of AR [156,157], with its protein expression escalating in the context of castration resistance and being associated with a poor prognosis. BET (bromodomain and extraterminal) inhibitors have exhibited antitumor activity in CRPC models, attenuating AR signaling and reducing c-MYC expression [158,159]. Moreover, BET inhibitors have been demonstrated to decrease AR-V7 levels by regulating the splicing factors required for AR-V7 generation [160].

### 4.1. DNA Methylation

The contribution of DNA methylation to PCa development was first reported in the early 1990s [161]. Approximately 22% of tumors in PCa are associated with hypermethylation [162]. DNA methylation is linked to gene silencing and occurs when a methyl group is added to the C5 position of the cytosine residues in CpG dinucleotides [163]. DNMT (DNA methyltransferase) enzymes catalyze the addition of 5-methyl cytosine to DNA, which can be removed by DNA demethylases from the ten-eleven translocation (TET) family [164]. In addition, approximately 60% of all gene promoters are associated with CpG islands [165]. Consequently, abnormal DNA hypermethylation at CpG islands can lead to gene silencing, including the inactivation of tumor-suppressor genes [47,166]. The downregulation of *TET2* has been implicated in the regulation of AR signaling and the development of PCa [167,168]. The diminished expression of both TET2 and 5-hydroxymethylcytosine, associated with PCa progression, has been identified as a potential prognostic biomarker [169,170].

Cytidine methylation of regulatory sequences near *GSTP1*, which encodes an enzyme responsible for safeguarding DNA against oxidants and carcinogens, has been associated with prostate tumorigenesis [171]. In PCa, *GSTP1* is methylated in its promoter region, leading to the diminished expression of GSTP1 in tumor cells. Hypermethylation of the *GSTP1* promoter occurs in approximately 75% of pre-invasive high-grade prostatic intraepithelial neoplasms and in over 90% of prostate tumors and persists through all stages of PCa progression [172]. In patients with localized disease, pre-operative circulating methylated *GSTP1* in plasma has been demonstrated to predict PSA recurrence and tumor aggressiveness [173,174,175].

Promoter hypermethylation can lead to the silencing of not only tumor-suppressor genes but also key receptors, such as AR and ESR1. Additionally, cell adhesion genes (such as *CD44* or *CDH1*), cell cycle genes (such as *CCND2*, *CDKN1B*, or *SFN*), and apoptosis-related genes (such as *ASC*, *BCL2*, *DAPK*, or *PTGS2*) can be affected by this process [176]. Although global hypomethylation is common in primary tumors, it becomes more pronounced in metastatic PCa [177]. Notably, 22% of mCRPC tumors exhibit distinct epigenomic features characterized by hypermethylation and the enrichment of somatic mutations involving *TET2*, *DNMT3B*, *IDH1*, and *BRAF* [162].

DNMT inhibitors such as azacytidine and decitabine have been developed to target aberrant DNA hypermethylation; furthermore, their efficacy against PCa has been clinically evaluated (NSC127716). In a phase 1/2 study of azacitidine in combination with docetaxel in mCRPC, a PSA response was observed in 52% (10/19) of patients, with no dose-limiting toxicity reported [178].

### 4.2. Histone Methylation

Histones undergo methylation when one, two, or three methyl groups are added from S-adenosylmethionine to the side chains of arginine, lysine, and histidine residues. Histone methylations, such as H3K4me1, H3K9me2, and H3K9me3, are reduced in PCa tissues when compared with their levels in normal prostate tissues [179]. However, H3K27me3 marks are strongly enriched in the promoter regions of tumor-suppressor genes in metastatic PCa when compared with those in localized PCa and normal prostate tissues [150,180]. 

The increased genomic distribution of H3K27me3 in metastatic PCa can primarily be attributed to the overexpression of EZH2 [154], a histone methyltransferase. EZH2 plays a pivotal role in promoting lineage plasticity and differentiation changes, which are closely associated with neuroendocrine PCa (NEPC) [181]. The polycomb group proteins EZH2 and embryonic ectoderm development (EED), another member of the PRC2 complex, are currently under investigation as therapeutic targets [182]. EED functions as a direct regulator of AR and its downstream genes, working in conjunction with EZH2 in the context of AR-positive PCa [183]. EZH2 represents a promising target, and several EZH2 inhibitors have been identified in early-stage clinical studies [184]. These EZH2 inhibitors are currently under evaluation in clinical trials, both as standalone treatments and in combination with ARSIs or immunotherapy for PCa [185].

Conversely, histone demethylases are enzymes responsible for removing methyl groups from histones. LSD1 (KDM1A), a multiple histone demethylase, is overexpressed in patients with advanced PCa [186]. LSD1, which specifically demethylates H3K4me1 and H3K4me2 [187], interacts with AR, activating AR-dependent transcription or a subset of cell cycle gene expression [188,189]. A small-molecule LSD1 inhibitor, SP2509, has been shown to attenuate tumor growth in CRPC xenograft models [190]. The inhibition of LSD1 disrupts the chromatin binding of FOXA1, a transcription factor for AR [191]. By regulating AR binding and gene expression, LSD1 inhibition not only suppresses tumor growth but also exhibits a synergistic interaction with enzalutamide in CRPC models [190]. These findings lend support to the use of LSD1 as a potential therapeutic target, with the LSD1 inhibitor INCB059872 currently undergoing clinical development. In addition, a clinical trial involving a novel LSD1 inhibitor, CC-90011, has recently been initiated [192].

### 4.3. Histone Acetylation

Histone acetylation involves the addition of an acetyl group to the lysine residues of histones, leading to the opening and activation of chromatin. Typically, this process is associated with the activation of transcription, whereas histone deacetylation is often linked to gene silencing [193]. Super-enhancers, which are clusters of enhancers marked by high H3K27ac levels, play a crucial role as drivers of oncogenic processes in cancer cells [194,195]. The activation of histone acetyltransferases, such as p300 and CREB-binding protein (CBP), is strongly associated with an increase in H3K27ac modification frequency [196]. Moreover, p300 and CBP play pivotal roles in the regulation of key genes, including AR target genes, in PCa [197]. Inhibitors targeting p300 and CBP, including CCS1477, A-485, and FT-7051, have been developed [198,199]. The role of histone deacetylase inhibitors in CRPC treatment has also been investigated. This exploration involves the downregulation of AR, AR splice variants, and c-Myc expression. Additionally, these inhibitors have been associated with loss-of-function mutations in p300 or CBP, driving synthetic lethality [197]. Clinical trials evaluating the effectiveness of CCS1477 and FT-7051 in treating PCa have commenced recently.

Histone deacetylases (HDACs) are enzymes responsible for removing histone acetylation; thus far, several types of HDACs have been identified in humans [200]. Notably, there are five classes of HDAC inhibitors, including hydroxamic acids, cyclic tetrapeptides, short-chain carboxylic acids, benzamides, and ketoderivatives [201]. HDAC overexpression has been observed in various malignancies, including PCa [202]. Notably, the expression of HDAC1 and HDAC2 is positively correlated with higher Gleason scores in PCa, whereas that of HDAC1/2/3 is positively associated with the proliferative marker Ki67 [203]. Concordantly, HDAC expression is linked to poor clinical outcomes, and HDAC inhibitors are regarded as potential therapeutic options [203]. Several HDAC inhibitors, such as vorinostat, pracinostat, panobinostat, and romidepsin, have undergone testing in phase 2 clinical trials for PCa [204,205]. However, most patients in these trials experienced either toxicity from these agents or disease progression [206]. Clinical trials involving HDAC inhibitors have not yielded significant success, mainly owing to issues such as poor oral bioavailability, lack of drug selectivity, and other mechanisms that require further exploration [206].

## 5. Non-Coding RNAs

### 5.1. Long Non-Coding RNAs (lncRNAs) and Micro-RNAs (miRNAs)

lncRNAs have been implicated in prostate carcinogenesis. They exert their influence through various mechanisms, including regulation of the AR expression pathway, epithelial–mesenchymal transition, miRNAs, and the PI3K/Akt/mTOR pathway; these processes are governed by PCa gene expression marker 1 (PCGEM1), PlncRNA-1, colon cancer-associated transcript 2 (CCAT2), C-terminal binding protein 1 antisense (CTBP1-AS), growth arrest-specific 5 (GAS5), long intergenic non-protein-coding RNA 1296 (LINC01296), LOC400891, and LOC440040, etc. [207,208,209,210,211,212,213,214,215,216]. 

LINC00624 is overexpressed in both PCa tissues and cells and is associated with malignant progression. Furthermore, the relationship between NF-κB and PCa is linked to the expression of LINC00624, as it forms a co-regulatory axis with TEX10, stimulating NF-κB activity. Therefore, LINC00624 is hypothesized to function as an oncogene in PCa progression [90]. The activation of AKT and NF-κB signaling involves a crucial regulatory protein complex comprising PHLPP, FKBP51, and IKKα. The lncRNA PCAT1 is implicated in the regulation of the PHLPP/FKBP51/IKKα complex, directly binding to FKBP51 and excluding PHLPP from the complex. This regulation by PCAT1 contributes to CRPC progression by activating AKT and NF-κB signals [91].

Upregulated AR expression is a hallmark of CRPC and may serve as a mechanism facilitating AR signaling in the presence or absence of low endogenous androgen levels. Specific lncRNAs, such as CTBP1-AS and HOTAIR, exhibit upregulation following ADT, signifying their negative regulation by androgens. This upregulation of lncRNAs results in increased AR expression. CTBP1-AS upregulation leads to the downregulation of CTBP1, which negatively modulates AR by inhibiting androgen-mediated demethylation. Consequently, CTBP1-AS-induced CTBP1 downregulation results in AR upregulation [207]. HOTAIR binds to the N-terminal domain of the AR, hindering the binding of the E3 ubiquitin ligase MDM2, which interacts with the AR through the same domain. This blockade prevents AR ubiquitination and degradation [217]. PCGEM1 and PRNCR1 bind to AR, enhancing AR-mediated gene activation processes. PCGEM1 is reportedly dependent on PRNCR1 for binding. The relationship between AR and PCGEM1/PRNCR1 was examined via in vitro experiments using androgen-sensitive LNCaP cells [218]. Both PCGEM1 and PRNCR1 were significantly upregulated in CRPC cell line models and activated AR in the absence of androgens. The knockdown of either PCGEM1 or PRNCR1 significantly inhibited in vivo tumor growth in a xenograft mouse model [209], but these proteins were not implicated in CRPC or AR signaling. PCGEM1, but not PRNCR1, was associated with PCa. Notably, neither PCGEM1 nor PRNCR1 were associated with poor clinical outcomes [212]. RNA-sequencing data from two distinct androgen-dependent models revealed that PCGEM1 was considerably expressed in PCa, whereas PRNCR1 was minimally expressed [212]. PCGEM1 was implicated in early-stage PCa but not in metastatic PCa, being significantly downregulated in metastatic tumors relative to primary tumors and significantly repressed in patients with poor outcomes [218]. Significantly reduced PCGEM1 expression and AR activity were also observed post-castration [209]. Conversely, the AR stimulation-induced upregulation of PCGEM1 expression was observed in vivo [218].

Interaction with miRNAs is another mechanism by which lncRNAs regulate AR activity. PlncRNA-1 likely protects AR from miRNA-mediated suppression [211] by acting as a sponge for certain AR-targeting miRNAs, including miR-34c and miR-297. PCGEM1 and HOTAIR are reportedly targeted by miR-34a [219] and miR-145 [210], respectively.

In PCa, both miR-145-5p (the guide strand) and miR-145-3p (the passenger strand) are downregulated. The introduction of these microRNAs into PCa cells reportedly suppresses their proliferative, migratory, and invasive capabilities. Furthermore, miR-145-3p regulates the expression of cancer-promoting genes such as *MELK*, *NCAPG*, *BUB1*, and *CDK1*. Patients with PCa who exhibit the expression of these genes are more commonly associated with CRPC than with CSPC [220].

The cancer-suppressive functionality of both guide and passenger strands of several microRNAs, including miR-30a, miR-139, miR-143, and miR-145, has been established through large-scale cohort analysis using The Cancer Genome Atlas [221]. The miRNA expression profile of PCa using needle biopsy samples from CSPC and normal prostate tissues revealed that miR-222 and miR-31 exhibit cancer-suppressive functions [222].

miR-130b exhibits significant downregulation among miRNAs in PCa tissues, and its expression is significantly reduced in PCa cell lines as well. The decreased expression of miR-130b significantly promotes the proliferation, invasion, and tubule formation of PCa cells, whereas its overexpression suppresses angiogenesis in PCa. miR-130b directly regulates TNF-α, inhibiting the NF-κB signaling pathway and its downstream gene, *VEGFA*. Furthermore, VEGFA, by reducing the expression of miR-130b, contributes to a feedback loop involving miR-130b/TNF-α/NF-κB/VEGFA that is associated with angiogenesis in PCa [89]. Moreover, elevated miR-130b levels are reportedly correlated with race disparity and PSA levels/failure in African-American patients. miR-130b is significantly overexpressed in this population when compared with its levels in tissue samples and cell lines from European-American patients. Elevated miR-130b levels reportedly act as an independent prognostic biomarker for African-American patients, highlighting their potential role as a risk factor for poor prognosis in such individuals with PCa. *FHIT* and *CDKN1B* are direct functional targets of miR-130b. Additionally, other miR-130b target genes, including *TP53INP2*, *PTEN*, *FOXO1*, *KDM2A*, *BTG1*, and *CDKN1A*, are upregulated in anti-miR-130b-stable cells when compared with those in controls [223].

In PCa, reports of liquid biopsy using upregulated or downregulated microRNAs in the blood (serum, plasma, exosomes) and urine as indicators have been increasing [224] [225]. Several studies suggest the detection of cell-free microRNAs, such as miR-141, miR-375, miR-107, and miR-221, in the blood as useful biomarkers for PCa [226]. Combining miR-17-3p and miR-1185-2-3p for PCa diagnosis has exhibited >90% sensitivity and specificity, highlighting its promise in clinical applications and circumventing the need for unnecessary prostate biopsies [227].

### 5.2. Role of miRNAs in PCa Drug Resistance

Growing evidence suggests that miRNAs play a pivotal role in anti-AR drug resistance. For example, miR-23b and miR-27b sensitize CRPC cells to flutamide by targeting CCNG1 [228]. miR-221 and miR-222, upregulated in CRPC cells, maintain resistance in PCa [229], and miR-663 affects castration resistance by modulating the AR signal [230]. In mCRPC, patients with higher levels of miR-375 and miR-3687 in whole blood exhibit a shorter time to progression with enzalutamide treatment [231]. Other miRNAs target AR and its splicing variant, AR-V7, to resensitize drug-resistant PCa cells. For instance, miR-212 inhibits hnRNPH1, thereby reducing AR and AR-V7 expression, whereas miR-361-3p enhances enzalutamide sensitivity by targeting AR-V7 [232,233,234]. In addition, miRNAs can serve as prognostic biomarkers in CRPC. High miR-1290 and miR-375 expression is associated with poor survival rates and helps to predict the CRPC stage [234].

Additionally, miRNAs play a key role in taxane resistance. Several miRNAs, including miR-34b-3p, miR-199a, miR-200b-3p, and miR-375, have been implicated in paclitaxel resistance [235,236]. Docetaxel resistance is associated with miR-375 and miR-323 upregulation, whereas miR-195, miR-204, miR-143, and miR-200b can enhance docetaxel sensitivity in PCa [237,238,239,240]. Furthermore, elevated levels of six plasma microRNAs (miR-132, miR-200a, miR-200b, miR-200c, miR-375, and miR-429) have been identified as predictors of decreased OS with docetaxel treatment [241]. Such findings indicate the potential of circulating miRNA expression in the blood as a complementary tool to PSA measurements both in diagnosis and during treatment and highlight the complex roles of miRNAs in the development of resistance to antiandrogen and taxane therapies in PCa.

In the context of PCa, there is an increasing research interest in developing a minimally invasive liquid biopsy using blood or urine to assess treatment progress. However, selecting appropriate miRNA indicators poses a major challenge. Notably, significant variation in miRNA levels among different research facilities is apparent when reviewing research on miRNA levels in urine from patients with PCa [242,243,244,245,246] (Table 3). This inconsistent pattern confounds the utilization of these miRNAs for universal diagnostics. Therefore, further research on the development of liquid biopsy for PCa and the standardization of specimen collection methods, RNA extraction techniques, and detection systems is warranted.

## 6. Prostate-Specific Membrane Antigen (PSMA)

PSMA is a type II transmembrane protein expressed in most high-grade, metastatic, androgen-insensitive, clinically significant cases of PCa. More than 90% of PCa cases exhibit PSMA overexpression at levels reported to be 100–1000 times higher than those in normal prostate tissue [247,248,249,250,251]; however, PSMA is scarcely expressed in benign or hyperplastic prostate tissue [252]. The organs reported to exhibit high PSMA expression, apart from the prostate, are the kidney, bladder, salivary glands, ganglion, liver, spleen, duodenum, and colon [253,254,255]. PSMA is a favorable target for molecular imaging [256,257,258,259,260,261,262] and therapy [263,264,265,266,267] in PCa. The US Food and Drug Administration (US FDA) has approved three types of radionuclides for positron emission tomography (PET) imaging of PSMA, namely F-18 piflufolastat PSMA, Ga-68 PSMA-11, and F-18 flotufolastat PSMA, for patients with PCa. PSMA PET-computed tomography (PET/CT) or magnetic resonance imaging (PET/MRI) are recommended for detecting metastases and restaging PCa in cases of biochemical recurrence [268]. In the Advanced Prostate Cancer Consensus Conference (APCCC 2022), 77% of panelists reached a consensus to perform PSMA PET/CT on the majority of patients with clinically localized high-risk PCa. However, 92% of panelists voted not to recommend PSMA PET/CT or MRI for patients with clinically localized favorable intermediate-risk (NCCN definition) PCa [269]. Determining superiority between ^68^Ga and ^18^F-PSMA PET/CT is challenging because their accumulation in normal organs is similar and their diagnostic accuracy does not differ significantly [260,261,262,270]. In clinical practice, a lack of high-level evidence remains in several aspects of PSMA PET/CT or MRI, and we look forward to future research contributing such evidence.

Radiation therapy utilizing the labeling of PSMA ligands to target PSMA-positive cells and the surrounding microenvironment involves both alpha and beta radionuclides such as ^225^Ac-PSMA and ^177^Lu-PSMA, respectively [255,271,272,273]. In phase 3 international collaborative non-blinded trial evaluating ^177^Lu-PSMA-617 in patients with mCRPC who had a treatment history involving one or more ARSI treatments and 1–2 taxane chemotherapy regimens and were PSMA-positive on PSMA PET/CT, ^177^Lu-PSMA-617 with standard treatment extended both the image-based PFS (median 8.7 vs. 3.4 months, hazard ratio 0.40, 99.2% CI 0.29–0.57, *p* < 0.001) and OS (median 15.3 vs. 11.3 months, hazard ratio 0.62, 95% CI 0.52–0.74, *p* < 0.001) significantly compared to those of standard treatment alone [263]. Therefore, ^177^Lu-labeled radiopharmaceutical ^177^Lu-PSMA-617 was approved by the US FDA and European Medicines Agency in 2022 as a last-line therapy for mCRPC. Currently, a ^225^Ac-labeled PSMA-targeting inhibitor is also being developed. Numerous clinical trials targeting localized CSPC (ClinicalTrials.gov number: NCT04430192, NCT04443062, and NCT05162573), mCSPC (Table 4), and CRPC (Table 5) are currently underway. 

## 7. Precision Medicine: Genome-Based Therapy

### 7.1. PARP Inhibitors (PARPis) for mCRPC with Alterations in HRR Genes

PARP recognizes single-strand DNA breaks and facilitates the transportation of base-excision repair proteins for genetic repair. Conversely, several proteins, including BRCA, play a pivotal role in HRR, particularly in the repair of double-strand DNA breaks. DNA repair can still occur in cases where a malfunction occurs in either PARP or BRCA. However, the administration of PARPis prevents DNA repair in cases where cancer cells exhibit BRCA abnormalities [96,275,276,277,278]. PARPis traps PARP-1 on damaged DNA, inhibiting auto-poly(ADP-ribosyl)ation and enhancing DNA affinity for the catalytic site, ultimately leading to cell death [96,279,280,281]. Notably, tumor cells with BRCA mutations are considerably more sensitive to PARPi treatment, exhibiting an approximately 1000-fold higher sensitivity than wild-type BRCA cells [282]. Among all HRR genes, the biallelic loss of *BRCA2* appears to yield the most significant benefit from PARP inhibition [283]. Durable responses to one class of treatment following progression to the other class appear to be infrequent [284], likely owing to shared resistance mechanisms, including *BRCA* reversion mutations [285]. Assuming that abnormalities in *ATM* and *BRCA1/2* confer increased sensitivity to PARPis, 19.3% of individuals with mCRPC may likely benefit from this therapy [122]. However, olaparib is ineffective in CRPC with *ATM* mutations [286].

PARPis, which were approved by the US FDA in 2023 and are used for monotherapy or combination therapy with other drugs for mCRPC, belong to four types: olaparib, niraparib, rucaparib, and talazoparib. Veliparib is currently undergoing clinical trials [287,288]. Preclinical studies have indicated variations in the ability of PARPi to trap PARP enzymes across different tumor cells, including PCa cells, with talazoparib exhibiting the highest PARP-trapping capacity among the tested inhibitors, followed by niraparib, olaparib, and rucaparib [289]. In all trials involving PARPis, both radiographic PFS (rPFS) and OS were longer in patients with HRR mutations, specifically *BRCA1/2* mutations, than in those without (Table 6) [288,290,291,292,293,294,295,296]. The TALAPRO-1 trial provided evidence of sustained antitumor activity of talazoparib in heavily pretreated patients with mCRPC and DDR-HRR mutations [297]. However, olaparib superiority was not observed in patients with alterations in other HRR genes in the PROfound trial (Table 2). Overall, although all PARPis share a similar mechanism of action, their differences in molecular structure and potency for trapping PARP enzymes may impact their clinical efficacy and potential use in different types of cancer.

Common adverse events reported with PARPi treatments include anemia and nausea. The incidence of anemia as an adverse event was similar across clinical trials of each PARPi, excluding veliparib (all-grade anemia: 46.0–65.8% and grade ≥ 3: 15.1–46.5%); however, the rate of anemia incidence was higher for talazoparib than for the others [288,291,292,293,294,295,296,298]. The rate of anemia incidence (all-grade and grade ≥ 3 anemia) due to veliparib was the lowest; however, an important consideration is that this trial had a small sample size (*n* = 74) [288]. The frequency of nausea as an adverse event varied across trials (all grade: 2.7–53.2% and grade ≥ 3: 0.5–4.2%); therefore, there may be specific characteristics associated with the occurrence of nausea due to PARPis (Table 7). Further research is required to elucidate the relationship between PARP-trapping potency and clinical toxicity profiles.

### 7.2. Akt Inhibitor for mCRPC with PTEN Deficiency

PTEN typically exerts tumor-suppressor effects by inhibiting the PI3K/Akt/mTOR signaling pathway. However, PTEN deficiency is present in approximately 50% of patients with mCRPC. Upon PTEN protein deficiency, signal transduction through PI3K and Akt occurs via genomic or non-genomic pathways, leading to the dysregulation of the cell cycle and cell proliferation, the enhancement of metabolic pathways such as those for glycolysis and protein translation, and the promotion of vascular neogenesis and cell survival pathways. These processes ultimately culminate in tumor formation and growth [299]. In patients with mCRPC, the PI3K/Akt/mTOR signaling pathway is likely activated as a mechanism for acquiring resistance to ARSIs [299]. Therefore, a treatment approach that concurrently inhibits the PI3K/Akt/mTOR and AR signaling pathways has been proposed to potentially improve the prognosis of mCRPC patients with PTEN protein deficiency [299,300].

Ipataseltib is a novel, cooperative, and selective ATP-competitive small-molecule inhibitor targeting all three isoforms of Akt [301]. A phase 3 trial (IPATentia150 trial: NCT03072238) evaluating the combination therapy of ipatasertib and apalutamide revealed that the median rPFS of mCRPC patients with PTEN deficiency was 18.5 months (95% CI: 16.3–2.1 months) in the ipatasertib + apalutamide group and 16.5 months (95% CI: 13.9–17.0 months) in the placebo + apalutamide group. This combination therapy significantly increased rPFS by 23% (HR: 0.77, 95% CI: 0.61–0.98, *p* = 0.0335) [302,303]. Akt inhibitors are considered potential novel agents representing the precision medicine era for mCRPC, and a phase 3 randomized CAPItello-280 trial (NCT05348577) is underway to evaluate the efficacy and safety of capivasertib and docetaxel administration as compared to that of placebo and docetaxel administration in the treatment of patients with mCRPC [304]. Furthermore, the randomized CAPItello-281 trial (NCT04493853) is underway to evaluate the efficacy of capivasertib, a potent selective inhibitor of all three Akt isoforms (AKT1/2/3), in combination with abiraterone in patients with mCSPC whose tumors are characterized by PTEN deficiency [305].

### 7.3. Immune Checkpoint Inhibitors (ICIs) for mCRPC

The programmed cell death 1 (PD-1)/programmed cell death ligand 1 (PD-L1) pathway is pivotal in regulating T-cell activity [306,307,308]. Several clinical studies have assessed the efficacy of ICIs, including anti-PD-L1 (nivolumab, atezolizumab, durvalumab, and avelumab), anti-PD1 (pembrolizumab and nivolumab), and anti-CTLA4 (ipilimumab) antibodies. Although preliminary studies have indicated limited anticancer activity of ICIs [309,310], patient selection based on deficiencies in *MMR* genes is considered crucial, as this patient cohort can potentially respond to ICIs. The FDA has approved the use of the anti-PD1 antibody pembrolizumab for treating cancers, including PCa with MMR mutations or MSI [311]. However, responses to ICIs in MMR-mutated or MSI PCa cases are not consistent.

Strong correlations were reported between immunohistochemical PD-L1 expression and the efficacy of PD-1 blockade in various malignant tumor cohorts. However, within these cohorts, only two patients had CRPC [312]; their samples revealed negative PD-L1 expression, suggesting the lack of a significant association. High PD-1 expression was observed in infiltrating T lymphocytes, specifically in CD8-positive prostate tissues, suggesting that they may not effectively mount an immune response [313]. A study involving 17 radical prostatectomy specimens investigated the expression of FoxP3, PD-1, and B7-H1 and revealed clusters of lymphocytes that were positive for all three proteins in the proximity of PCa lesions, collectively contributing to the negative regulation of the tumor immune response [139].

In one study, over 50% of patients with CRPC-associated, MSI-high tumors, or MMR mutants achieved a 50% reduction in PSA level after pembrolizumab treatment [118]. However, the rationale behind the lack of response to ICI therapy in some patients with PCa and MMR loss/MSI-high status remains unclear. Nonetheless, interest in combining ICIs with other therapies remains high. Concurrently, promising results were observed from a phase 1/2 clinical trial of pembrolizumab plus docetaxel, ARSIs, or PARP inhibitors in patients with mCRPC [314]. Furthermore, several phase 3 clinical trials evaluating the efficacy of pembrolizumab combined with docetaxel, enzalutamide, and olaparib are ongoing.

A phase 1/2 non-randomized trial (CA184-107) targeted patients with mCRPC who had not undergone more than one regimen of chemotherapy. In this trial, ipilimumab was administered as a single agent or in combination with external beam radiation therapy. The effectiveness of ipilimumab as monotherapy was reportedly between 14.3% and 16% in terms of the PSA50% response rate [315]. In another phase 3 randomized clinical trial (CA184-043) conducted with patients with mCRPC and bone metastases progressing after docetaxel chemotherapy, a single fraction of radiation therapy was administered to target bone metastases. Following this, patients were randomly assigned to receive either ipilimumab or a placebo [316]. While no significant difference in OS was observed between the ipilimumab and placebo groups (11.2 vs. 10 months, *p* = 0.053), the PFS was notably improved in the ipilimumab group compared with that in the placebo group (4 vs. 3.1 months, HR 0.70, *p* < 0.0001). Furthermore, the PSA response rate was significantly higher in the ipilimumab group than in the placebo group (13.1 vs. 5.2%). These findings highlight the potential benefits of ipilimumab in the treatment of patients with mCRPC and bone metastases following docetaxel chemotherapy [317,318]. Immunotherapy with ipilimumab is expected to be more effective in earlier stages of CRPC with minimal disease progression. New trials involving ipilimumab, either as a standalone treatment or in combination, are currently underway in chemotherapy-naïve PCa (NCT01057810 CA184-095) and neoadjuvant settings (NCT01194271) as phase 3 trials.

Notably, up to 10% of patients with mCRPC present with *CDK12* aberration, which is associated with the response to ICIs [95]. CDK12, which forms a complex with cyclin K, is critical for DNA repair during gene translation; consequently, its inactivation leads to focal tandem duplications that promote gene fusions or mutations, thus enhancing neoantigen production and the tumor immune response [95]. One-half (2/4) of patients with mCRPC-causing *CDK12* mutations showed PSA responses after administration of a PD1 inhibitor [95]. A phase 2 clinical trial to evaluate the efficacy of ipilimumab and nivolumab for *CDK12*-mutated mCRPC is ongoing [319]. Notably, *CDK12* is mutated in 5% of patients with mCSPC and has been associated with improved responses to ICIs [121,320,321].

Defective MMR is associated with a higher tumor grade, a greater likelihood of developing de novo metastatic PCa, and a poorer response to ARSIs [322]. The identification of MMR-deficient PCa may have immediate clinical relevance considering recent data suggesting the sensitivity of such tumors to immune checkpoint blockade in other malignancies [323,324,325,326]. Favorable responses to ICIs such as pembrolizumab have been reported in MMR-deficient, MMR-mutated, MSI-high, or high-tumor-mutational-burden mCRPC [327,328,329,330,331,332,333]. However, prospective trials comparing ICIs to standard systemic therapy in MMR-deficient PCa are currently lacking [334,335].

An examination of PD-L1 expression in PCa cells and PD-1 expression in TILs among 16 patients with CRPC revealed high PD-1/PD-L1 expression in three patients. However, no TIL infiltration was observed in four cases, and PD-L1 expression was not detected in four of the eight cases [336]. The examination of PD-L1 expression (using clone EPR1161) in 873 hormone therapy-naïve radical prostatectomy specimens confirmed its expression in over 50% of the cases. Furthermore, high PD-L1 expression was significantly associated with cell proliferation, as evidenced by Ki-67, AR expression, and, notably, high PSA recurrence rates [337]. Collectively, poorer prognosis in cases with high PD-L1 expression in hormone therapy-naïve patients indicates the potential for improving outcomes in patients with PCa at a higher risk of recurrence, even before reaching the CRPC stage, by targeting the tumor immune response through PD-1/PD-L1 control. The analysis of blood samples and enzalutamide-resistant PCa cell lines in patients with CRPC who developed resistance to enzalutamide revealed increased numbers of PD-L1/L2-positive circulating dendritic cells in the bloodstream of enzalutamide-resistant patients with CRPC. Additionally, a pronounced increase in PD-L1 expression was observed in enzalutamide-resistant PCa cell lines derived from LNCap [338]. Many monotherapies and combination therapies with anti-PD-1, anti-PD-L1, and anti-CTLA-4 antibodies have not shown promising therapeutic effects or gained approval for PCa; nonetheless, significant effectiveness has been observed in some cases. Therefore, establishing biomarkers that can predict such effectiveness could help establish these therapies as a promising option. Additionally, combination therapy trials using ICIs and various treatments are currently underway and hold the promise of establishing effective treatment methods (Table 8, Table 9 and Table 10).

## 8. Conclusions

PCa is characterized by its androgen dependency, with treatment primarily based on androgen signal deprivation. However, the involvement of driver genes in both the initiation and progression of PCa is highly diverse among cases. Precision medicine, which involves understanding individual genetic alterations and proposing tailored treatment strategies, is considered crucial in the context of PCa.

The development of treatments based on genetic mutations such as PARPis for HRR gene mutations, AKT inhibitors for patients with genetic mutations in the PI3K/Akt pathway, and ICI for mutations in MMR genes and cases of MSI-high or high-tumor-mutational-burden CSPC and CRPC is anticipated. Monitoring the treatment progress of patients with PCa using liquid biopsy methods such as miRNA in blood or urine is a promising field that could significantly impact routine PCa treatment.

## Figures and Tables

**Table 1 cancers-16-00523-t001:** Frequency of genetic alterations in each stage of prostate cancer.

Reference	Stage	Gene									
		*AR*	*TP53*	*PTEN*	*RB1*	*SPOP*	*BRCA2*	*BRCA1*	*ATM*	*CDK12*	*ETS*
[11]	Localized	0	8	17	1	11	3	0.3	6	1	54
[27]	Localized	0	36.3	54.5	18.2	0	9.1	0	9.1	0	72.7
[20]	mCSPC	5	33	24	5	13	7	0.2	5	6	
[10]	mCRPC	62.7	53.3	40	9.3	8	13.3	0.7	7.3	4.7	56.7
[28]	Localized	2	27	12	2	12	6	1	2	4	
	mCSPC	4	30	18	7	11	7	1	2	6	
	mCRPC	52	48	29	18	5	10	2	11	11	
[29]	mCSPC	0–5	30–48	12–34	5–8	7–11	2–9	1–2	3–6	4–6	
	mCRPC	41–52	30–50	32–40	6–13	6–11	7–12	1–2	5–6	4–6	
[30]	Localized	1.2	8	17	0.9	12	3	1.2	7	2.4	46
	mCSPC	6	33	25	5	13	7	0.5	8	7	26
	mCRPC	59	40	33	13	11	12	5	8	10	32

*AR*, androgen receptor; *TP53*, tumor protein 53 (p53); *PTEN*, phosphatase and tensin homolog; *RB1*, retinoblastoma protein 1; *SPOP*, speckle-type POZ protein; *BRCA2*, breast cancer 2 gene; *BRCA1*, breast cancer 1 gene; *ATM*, ataxia-telangiectasia mutated; *CDK12*, cyclin-dependent kinase 12; *ETS*, E26 transformation-specific transcription factor; mCSPC, metastatic castration-sensitive prostate cancer; mCRPC, metastatic castration-resistant prostate cancer.

**Table 2 cancers-16-00523-t002:** Overall survival for patients with single gene mutations in HRR-related genes.

Gene	Treatment	*n*	Median OS (Months)	HR (95% CI)
*BRCA2*	Olaparib	81	24.8	0.59 (0.37–0.95)
	ARSI	47	15.2	
*CDK12*	Olaparib	61	14.1	0.97 (0.57–1.71)
	ARSI	28	11.5	
*ATM*	Olaparib	62	18.0	0.93 (0.53–1.75)
	ARSI	24	15.6	
*BRCA1*	Olaparib	8	11.7	0.42 (0.12–1.53)
	ARSI	5	9.4	
*CHECK2*	Olaparib	7	16.6	0.87 (0.19–4.44)
	ARSI	5	17.1	
*PPP2R2A*	Olaparib	6	8.1	5.11 (1.10–35.73)
	ARSI	4	NE	
*RAD51B*	Olaparib	4	NE	NE
	ARSI	1	3.6	
*RAD54L*	Olaparib	3	19.3	NE
	ARSI	2	5.7	

ARSI, androgen receptor signaling inhibitor; CI, confidence interval; HR, hazard ratio; HRR, homologous recombination repair; NE, not estimable; OS, overall survival.

**Table 3 cancers-16-00523-t003:** Liquid biopsy using microRNAs in the urine of patients with prostate cancer.

		Number of Samples (Normal/PCa)		Altered miRNA Expression in Patients with PCa or a High Risk of PCa	
Reference	Samples	Screening	Validation	Upregulated miRNAs	Downregulated miRNAs
[193]	DRE-urinary sediments	10 PCa + 99 (50 high-risk vs. 49 low-risk)	40 (11 high-risk vs. 29 low-risk)	miR-3195, let-7b-5p, miR-144-3p, miR-451a, miR 148a-3p, miR-512- 5p, miR-431-5p	
[194]	Urinary exosomes	29 (9/20)	47 (19/28)		miR-196a-5p, miR-34a-5p, miR-501-3p, miR-92a-1-5p, miR-143-3p
[195]	Urinary exosomes	208 (20/188)	217 (20/197)	miR-30c-5p, miR-30b-5p, miR-30a-5p, miR-30a-3p, miR-125b-5p	miR-375, miR-21-5p, miR-141-3p, let-7c-5p, miR-26b-5p, miR-101-3p, let-7b-5p
[196]	DRE-urinary extracellular vesicles (EVs)	13 (4/9)	74 (26/48)	miR-10a-5p, miR-204-5p, miR-30a-3p	miR-27a, miR-328, miR-483, miR-125a, miR-146a, miR-195, miR-145, miR-150

DRE, digital rectal examination; PCa, prostate cancer.

**Table 4 cancers-16-00523-t004:** Clinical trials on PSMA-targeting radionuclide therapy for patients with mCSPC.

Drug	Concomitant Drug and/or Treatment	Phase	Number of Patients	Prior Treatment	Primary Endpoint	Identifier
^177^Lu-DOTA-TLX591	EBRT	2	50	RP	PSA-PFS	NCT05146973
^177^Lu-PSMA-617	SBRT	1	6			NCT05079698
^177^Lu-PSMA-617	ADT	3	1126		rPFS	NCT04720157
^177^Lu-PSMA-617	Docetaxel	2	140		Undetectable PSA rate	NCT04343885

ADT, androgen deprivation therapy; EBRT, external beam radiotherapy; mCSPC, metastatic castration-sensitive prostate cancer; PFS, progression-free survival; PSA, prostate-specific antigen; PSMA, prostate-specific membrane antigen; RP, radical prostatectomy; rPFS, radiographic PFS; SBRT, stereotactic body radiotherapy.

**Table 5 cancers-16-00523-t005:** Clinical trials on PSMA-targeting radionuclide therapy for patients with CRPC.

Drug	Concomitant Drug and/or Treatment	Phase	Number of Patients	Prior Treatment	Primary Endpoint	Identifier
^225^Ac-PSMA-I&T		2	100	Taxane		NCT05219500
^225^Ac-Labeled Antibody Targeting hk2		1	70	ARSI ± taxane		NCT04644770
^177^Lu-PSMA-617	±ADT	3	450	ARSI	rPFS	NCT04689828
^177^Lu-PSMA-I&T		3	400	ARSI	rPFS	NCT05204927
^177^Lu-PSMA-PNT2002		3	415	ARSI	rPFS	NCT04647526
^177^Lu-DOTA-rosopatamab		3	387	Taxane	rPFS	NCT04876651
^177^Lu-PSMA-617	Abiraterone		58	Taxane		[274]
^177^Lu-PSMA-617	Cabazitaxel	½	44	ARSI + docetaxel		NCT05340374
^177^Lu-PSMA-617	Enzalutamide	2	160	ADT	PSA-PFS	NCT04419402
^177^Lu-PSMA-617	Olaparib	1	52	ARSI + taxane		NCT03874884
^177^Lu-PSMA-617	Pembrolizumab	½	37	ARSI	PSA response	NCT03658447
^177^Lu-PSMA-617	Pembrolizumab	1	43	ARSI		NCT03805594
225Ac-J591	Pembrolizumab	1/2	76	ADT		NCT04946370
^225^Ac-PSMA-617	±^177^Lu-PSMA-617/I&T and/or ARSI	1	30	±^177^Lu-PSMA-617 and/or ARSI		NCT04597411
^225^Ac-J591 + ^177^Lu-PSMA-I&T		½	33	ARSI and taxane		NCT04886986

ADT, androgen deprivation therapy; ARSI, androgen receptor signaling inhibitor; CRPC, castration-resistant prostate cancer; hk2, human kallikrein-2; I&T, imaging and therapy; PFS, progression-free survival; PSA, prostate specific antigen; PSMA, prostate-specific membrane antigen; rPFS, radiographic PFS.

**Table 6 cancers-16-00523-t006:** Efficacy of PARPis, as monotherapy or combination therapy, in each clinical trial.

Reference	Comparator	Control	Participants	HR for rPFS	HR for OS
[290,291]	OLA	ARSI	BRCAm or ATMm	0.34 (0.25–0.47)	0.69 (0.50–0.97)
[292]	OLA + ABI	ABI	HRRm	0.45 (0.31–0.65)	Not shown
[293]	OLA + PEM	ARSI	BRCAm	0.40 (0.21–0.79)	0.52 (0.27–0.99)
[294]	NIRA + ABI	ABI	BRCAm	0.53 (0.36–0.79)	Not shown
[295]	RUCA	DOC or ARSI	BRCAm	0.50 (0.36–0.69)	0.81 (0.58–1.12)
[296]	TALA + ENZ	ENZ	HRRm	0.46 (0.30–0.70)	Not shown
[288]	VELI + ABI	ABI	DDRd	Not estimated	Not estimated

ABI, abiraterone; ARSI, androgen receptor signaling inhibitor; ATMm, Participants with tumors harboring aberrations in *ATM*; BRCAm, Participants with tumors harboring aberrations in *BRCA1/2*; DDRd, Participants with tumors harboring DNA damage response deficiency; DOC, docetaxel; ENZ, enzalutamide; HR, hazard ratio; HRR, homologous recombination repair genes; HRRm, Participants with tumors harboring aberrations in HRR genes; NIRA, niraparib; OLA, olaparib; OS, overall survival; PEM, pembrolizumab; rPFS, radiographic progression-free survival; RUCA, rucaparib; TALA, talazoparib; VELI, veliparib.

**Table 7 cancers-16-00523-t007:** Adverse events associated with PARPis, used as monotherapy or combination therapy in each clinical trial.

	Treatment		Anemia		Nausea	
	Comparator	Control	Active Drug	Control	Active Drug	Control
[291,298]	OLA	ARSI	49.6 (22.7)	15.4 (5.4)	43.0 (1.6)	20.8 (0)
[292]	OLA + ABI	ABI	46.0 (15.1)	28.1 (0.3)	16.4 (3.3)	12.6 (0.3)
[293]	OLA + PEM	ARSI	46.0 (19.6)	35.6 (1.5)	2.7 (1.2)	5.1 (0)
[294]	NIRA + ABI	ABI	46.2 (29.7)	20.4 (7.6)	23.6 (0.5)	13.7 (0)
[295]	RUCA	DOC or ARSI	46.7 (23.7)	17.7 (0.8)	49.6 (2.6)	19.2 (0.8)
[296]	TALA + ENZ	ENZ	65.8 (46.5)	20.6 (0.5)	17.5 (4.2)	12.5 (0.7)
[288]	VELI + ABI	ABI	20.3 (2.5)	14.9 (1.4)	53.2 (1.3)	6.8 (0)

(%), All-grade (Grade ≥ 3). ABI, abiraterone; ARSI, androgen receptor signaling inhibitor; DOC, docetaxel; ENZ, enzalutamide; NIRA, niraparib; OLA, olaparib; PEM, pembrolizumab; RUCA, rucaparib; TALA, talazoparib; VELI, veliparib.

**Table 8 cancers-16-00523-t008:** Clinical trials on ICIs in patients with mCSPC.

Drugs	Concomitant Drug and/or Treatment	Phase	Number of Patients	Patient Characteristics	Primary Endpoint	Identifier
Nivolumab or ipilimumab	Docetaxel	2/3	135	High-volume mCSPC	OS	NCT03879122 (PROSTRATEGY)
Pembrolizumab	Enzalutamide	3	1232	mCSPC	rPFS OS	NCT04191096 (KEYNOTE-991)
Nivolumab	Docetaxel	2	60	mCSPC with DRD or inflamed tumors	PSA response	NCT04126070
Nivolumab	Cabozantinib + abiraterone acetate	1b	22	De novo or recurrent mCSPC	Dose-limiting toxicities	NCT04477512 (CABIOS)
Cemiplimab	Docetaxel	2	20	De novo mCSPC	PSA response	NCT03951831 (REGN2810)
Durvalumab	SBRT	2	96	mCSPC with Oligomets	PFS	NCT03795207 (POSTCARD)

CSPC, castration-sensitive prostate cancer; OS, overall survival; rPFS, radiographic progression-free survival; PSA, prostate-specific antigen; DRD, DNA repair deficiency; NCT, ClinicalTrials.gov identifier.

**Table 9 cancers-16-00523-t009:** Clinical trials on ICIs in patients with mCRPC.

Drugs	Phase	Number of Patients	Prior Treatment	Patient Characteristics	Primary Endpoint	Identifier
Ipilimumab	3	988	Docetaxel		OS	NCT00861614 (CA184-043)
Ipilimumab	3	837	Chemotherapy-naïve		OS	NCT01057810 (CA184-095)
Atezolizumab	1b	35	Sipuleucel-T or enzalutamide		Safety	NCT01375842 (PCD4989g)
Nivolumab	2	38	Docetaxel	Germline and somatic DRD	PSA response	NCT03040791 (ImmunoProst trial)
Nivolumab	2	15	Local therapy	MMD, MSIh, TMBh, CDK12 inactivation	PSA response	NCT04019964
Pembrolizumab	2	30	ARSI	MMD, or CDK12 inactivation	PSA response ORR	NCT04104893 (CHOMP trial)
Pembrolizumab	2	50	ARSI	DRD	rPFS	NCT03248570
Pembrolizumab	2	258	Docetaxel and one or more ARSIs		ORR	NCT02787005 (KEYNOTE-199)
Pembrolizumab	2	100	Any	High mutational load, DRD, and/or MMD	PSA response ORR	NCT03506997 (PERSEUS1)
Nivolumab + ipilimumab	2	259	Docetaxel		ORR rPFS	NCT02985957 (CheckMate 650)
Nivolumab + ipilimumab	2	175	Systemic therapy	DRD or MMD or TMBh	PSA response rPFS CTC count	NCT03061539 (Neptunes)
Nivolumab + ipilimumab	2	75	ADT	MMD and/or TMBh	Disease control rate	NCT04717154 (INSPIRE)
Nivolumab + ipilimumab	2	40	Systemic therapy or radiation therapy	CDK12 inactivation	PSA response ORR	NCT03570619 (IMPACT)
Nivolumab + ipilimumab	2	15	Any	AR-V7	PSA response	NCT02601014 (STARVE-PC)

mCRPC, metastatic castration-resistant prostate cancer; OS, overall survival; NCT, ClinicalTrials.gov identifier; ARSI, androgen receptor signaling inhibitor; DRD, DNA repair deficiency; PSA, prostate-specific antigen; MMD, mismatch repair deficiency; MSIh, microsatellite instability-high; TMBh, tumor mutational burden-high; ORR, overall response rate; rPFS, radiographic progression-free survival; CTC, circulating tumor cell; ADT, androgen deprivation therapy; AR-V7, androgen receptor variant 7.

**Table 10 cancers-16-00523-t010:** Clinical trials on combination therapy with ICIs in patients with mCRPC.

Drugs	Concomitant Drugs and/or Treatment	Phase	Number of Patients	Prior Treatment	Primary Endpoint	Identifier
Atezolizumab	Sipuleucel-T	1b	37		Safety	NCT03024216
Atezolizumab	Radium-223	1b	44	ARSI	Safety ORR	NCT02814669
Atezolizumab	Cabozantinib	3	580	ARSI	PFS OS	NCT04446117 (CONTACT-02)
Atezolizumab	Enzalutamide	3	759	Abiraterone	OS	NCT03016312 (IMbassador250)
Durvalumab	Olaparib	2	17	ARSI	rPFS PSA response	NCT02484404
Durvalumab	Tremelimumab	2	52	ARSI	ORR	NCT02788773
Nivolumab	Docetaxel	2	84	ADT + ARSI	ORR PSA response	NCT03338790 (CheckMate 9KD)
Nivolumab	Pan-VEGFR/TIE2 tyrosine kinase inhibitor	2	49	ARSI	PSA response Adverse events	NCT04159896
Pembrolizumab	Enzalutamide	3	1200	Abiraterone-naïve CRPC or abiraterone for CSPC	OS rPFS	NCT03834493 (KEYNOTE-641)
Pembrolizumab	Docetaxel	3	1030	Chemotherapy-naïve	OS rPFS	NCT03834506 (KEYNOTE-921)
Pembrolizumab	Olaparib	3	529	Docetaxel + ARSI	OS rPFS	NCT03834519 (KEYLYNK-010)
Pembrolizumab	^177^Lu-PSMA-617	1b	37		Safety PSA response	NCT03658447 (PRINCE)
Nivolumab + ipilimumab	^177^Lu-PSMA-617	2	110	Docetaxel	PSA response	NCT05150236

mCRPC, metastatic castration-resistant prostate cancer; ARSI, androgen receptor signaling inhibitor; PFS, progression-free survival; OS, overall survival; NCT, ClinicalTrials.gov identifier; ADT, androgen deprivation therapy; ORR, overall response rate; PSA, prostate-specific antigen.

## Data Availability

Data can be found within the article.

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
