# Peer review of "Molecular Mechanisms of Prostate Cancer Development in the Precision Medicine Era: A Comprehensive Review"

_cancers, 2024, doi:10.3390/cancers16030523_

Round 1

Reviewer 1 Report

Comments and Suggestions for Authors

This review comprehensively describes the various mechanisms of resistance to ADT in prostate cancer and the multi-omic and epigenetic possibilities for preventing ADT resistance. I think this is a qualified review and I recommend accepting it.

Author Response

Thank you for the polite review.

I appreciate your consideration and judgment towards us.

Reviewer 2 Report

Comments and Suggestions for Authors Comments to authors: I have several reservations: Title: … in the precision medicine era…: Today prostate cancer oncology is interested in PSMA theranostics. Here expression of PSMA can lead to targeted radioligand therapy. FDA has approved three diagnostic tools and a radioligand. This is not part of the review. Abstract line 28: In this systematic review… The review is a narrative review and does not follow PRISMA guidelines. The review refers to literature of other cancers, but could also refer to previous reviews on the raised issues. For epigenetics, the review could be concerned of the added value relative to previous reviews [1, 2]. The review mainly cites ongoing clinical research. Of PARP inhibitors the review only points to olaparib. PCGW3 points to a sequence of phases for systemic treatments, and today the trends more for combination therapy with doublets or triplets than for monotherapy. bibliography [1] Kukkonen K, Taavitsainen S, Huhtala L, Uusi-Makela J, Granberg KJ, Nykter M, et al. Chromatin and Epigenetic Dysregulation of Prostate Cancer Development, Progression, and Therapeutic Response. Cancers (Basel) 2021;13. [2] Conteduca V, Hess J, Yamada Y, Ku SY, Beltran H. Epigenetics in prostate cancer: clinical implications. Transl Androl Urol 2021;10:3104-16.

Author Response

Comments from Reviewer 2

Our responses

I have several reservations: Title: … in the precision medicine era…: Today prostate cancer oncology is interested in PSMA theranostics. Here expression of PSMA can lead to targeted radioligand therapy. FDA has approved three diagnostic tools and a radioligand. This is not part of the review.

I appreciate your feedback and comments.

I have added a paragraph 6 about PSMA, providing details on diagnosis and treatment on pages 13-16.

Abstract line 28: In this systematic review… The review is a narrative review and does not follow PRISMA guidelines.

Thank you for your suggestion.

I have repraced the word at line 28.
In this comprehensive review, …

The review refers to literature of other cancers, but could also refer to previous reviews on the raised issues. For epigenetics, the review could be concerned of the added value relative to previous reviews [1, 2].

[1] Kukkonen K, et al. Cancers (Basel) 2021;13. [2] Conteduca V, et al. Transl Androl Urol 2021;10:3104-16.

I appreciate your thoughtful advice.

Following your guidance, I have incorporated references to several articles, including the ones you provided, and have expanded on the topic of epigenetics in more detail on page 8-10.

The review mainly cites ongoing clinical research. Of PARP inhibitors the review only points to olaparib. PCGW3 points to a sequence of phases for systemic treatments, and today the trends more for combination therapy with doublets or triplets than for monotherapy.

Thank you for the precise indication.

In the paragraph, 7.1. PARP inhibitor for mCRPC with alterations in HRR genes, I have provided detailed additions regarding the monotherapy and combination therapy of PARPi.

Reviewer 3 Report

Comments and Suggestions for Authors

This is a very well organized manuscript. The authors provided a nice overview on the Molecular Mechanisms of Prostate Cancer Development.  In my opinion they just need to include a short reference regarding the involvment of the NFKB pathway within the PCa development and progression (eg. Cells. 2022 Jul 5;11(13):2118. doi: 10.3390/cells11132118.) . This is due mainly to the increased number of paper that recognized a key role for this pathway into the PCa pathogenesis and also because several MicroRNA cited by the author are linked to NFkB.

Furthermore a graphical abstract could help readers. 

Author Response

Comments from Reviewer 3

Our responses

This is a very well organized manuscript. The authors provided a nice overview on the Molecular Mechanisms of Prostate Cancer Development.  In my opinion they just need to include a short reference regarding the involvment of the NFKB pathway within the PCa development and progression (eg. Cells. 2022 Jul 5;11(13):2118. doi: 10.3390/cells11132118.) . This is due mainly to the increased number of paper that recognized a key role for this pathway into the PCa pathogenesis and also because several MicroRNA cited by the author are linked to NFkB.

I appreciate the advice.

I have created a paragraph titled "2.6. NF-κB (nuclear factor-kappa B) Signaling" on page 5, providing a description of NF-κB. Additionally, I have added information about NF-κB within the paragraph on "5. Non-coding RNAs."

Furthermore a graphical abstract could help readers.

I have created a graphical abstract following your advice.

Round 2

Reviewer 2 Report

Comments and Suggestions for Authors In lines 596 you mention PSMA but may mean PSMA PET/CT. I did not find NCT04689828.

Author Response

Thank you for your careful review of our manuscript.

We revised our manuscript according to the comments from the deputy editor.

We underlined in revised points based on your indication. Would you please confirm them?

Comments from Reviewer 2

Our responses

In lines 596 you mention PSMA but may mean PSMA PET/CT.

I appreciate your feedback and comments. But I couldn't identify the points you mentioned; I apologize for any confusion. Instead, I changed 'PSMA PET' to 'PET/CT or MRI' in lines 599 and 603.

I did not find NCT04689828.

Thank you for your suggestion.

I have written based on the information from the following website.

“https://clinicaltrials.gov/study/NCT04689828”

“https://classic.clinicaltrials.gov/ct2/show/NCT04689828”

Woul you please review them? In your opinion, would it be better to delete this?

I am not qualified to assess the quality of English in this paper.

I appreciate receiving your honest feedback.

I have undergone multiple rounds of proofreading by the editing company Editage, and it has been checked by native speakers. I would appreciate it if you could point out specific areas of improvement in my inexperienced writing. Please let me know them, and I will convey the feedback to Editage. Thank you.